# Correlation between Obesity and Socioeconomic and Psychological Characteristics of Students Attending Different Rural School Types

**DOI:** 10.3390/children11060648

**Published:** 2024-05-27

**Authors:** Stephan Gretschel, Annabell Morgner, Cornelia Schindler, Nina Amelie Zierenberg, Henry Kusian, Meike Herkner, Stefan Reinsch, Frank Schoeneich, Edmund A. M. Neugebauer, Ulf Elbelt

**Affiliations:** 1Department of General, Visceral, Thoracic and Vascular Surgery, University Hospital Ruppin-Brandenburg, 16816 Neuruppin, Germany; h.kusian@ukrb.de; 2Brandenburg Medical School Theodor Fontane (MHB), 16816 Neuruppin, Germany; annabell.morgner@mhb-fontane.de (A.M.); cornelia.schindler@mhb-fontane.de (C.S.); nina.zierenberg@mhb-fontane.de (N.A.Z.); m.herkner@ukrb.de (M.H.); edmund.neugebauer@mhb-fontane.de (E.A.M.N.); 3Faculty of Health Sciences Brandenburg (FGW), University Hospital Ruppin-Brandenburg (UKRB), Fehrbelliner Straße 38, 16816 Neuruppin, Germany; stefan.reinsch@mhb-fontane.de; 4Centre for Health Services Research Brandenburg (ZVF-BB), 15562 Rüdersdorf, Germany; 5Department of Psychosomatics, 16816 Neuruppin, Germany; f.schoeneich@ukrb.de; 6Medical Clinic B, 16816 Neuruppin, Germany; ulf.elbelt@mhb-fontane.de; 7Division of Medicine, Department of Gastroenterology, Metabolism and Oncology, University Hospital Ruppin-Brandenburg, Brandenburg Medical School, 16816 Neuruppin, Germany

**Keywords:** obesity, schoolchildren, social determinants, psychological effects

## Abstract

We examined the prevalence of obesity in two types of schools—a comprehensive school and a grammar school—in a rural German region of Brandenburg. Methods: In a cross-sectional study, BMI values were measured in 114 students in grades 5, 7, and 10. In addition to the demographic data, data on nutrition, physical activity, and mental well-being were collected using a questionnaire. Results: A total of 44% (11/25) of the comprehensive school students and 15% (13/89) of the high school students are overweight, and 24% (6/25) of the comprehensive school pupils and 6% (5/89) of the grammar school pupils (*p* = 0.009) are obese. In addition, 91% (10/11) of the students with obesity, 36% (4/11) of the students with pre-obesity, and 31% (26/84) of the normal-weight students (*p* = 0.001) are concerned about their weight. Among the children with obesity, 82% (9/11) are afraid of gaining weight. In addition, 6% (5/82) of the normal-weight students, 25% (3/12) of the students with pre-obesity, and 70% (7/10) of the students with obesity feel restricted by their weight when exercising. Conclusion: School attendance and parental socioeconomic status appear to correlate with students’ weight statuses. There is a high level of suffering, and they feel uncomfortable with their bodies and worry about weight regulation.

## 1. Introduction

The prevalence of overweight (pre-obese or obese) is increasing worldwide in childhood and adolescence. The effects of overweight on psychological and physical development are not yet completely predictable, but there is evidence that they are serious [1]. Compared to normal-weight children of the same age, children with overweight have an increased risk of cardiovascular disease, lipid metabolism disorders, and high blood pressure. Being overweight can cause hormonal changes that lead to an earlier onset of puberty. Moreover, the likelihood that diseases such as type 2 diabetes mellitus, high blood pressure, and cardiovascular disease will manifest in adulthood is significantly increased by obesity in childhood and adolescence [2,3,4].

In addition to the physical diseases caused by obesity, 39% of children and adolescents with obesity also have mental disorders requiring treatment. These disorders have an unfavorable effect on self-esteem and quality of life. These mental disorders include anxiety disorders, depression, and social phobia. The risk of psychological disturbances is increased by 1.4-fold (OR = 1.40; 95% CI = 1.08–1.83) for adolescents with overweight and by 2.5-fold (OR = 2.50; 95% CI = 1.95–3.12) for adolescents with obesity [5]. Stigmatization leads to negative emotions, dissatisfaction, and the attribution of a negative stereotype of oneself in affected children, leading to their own increased devaluation and reduced self-esteem [6,7,8,9,10,11]. Zorn describes the situation of these adolescents as follows: “In reality, the young people affected are badly off. Their quality of life is lower than that of cancer patients: “They are often school dropouts and addicted to the internet, have no chance to get an apprenticeship or a job due to corpulence and are socially isolated” [12].

The nationwide KiGGS (German Child and Youth Health Survey) is a study that focuses on the health of children, adolescents, and young adults at the Robert Koch Institute in Germany, covering the years of 2014 to 2017. It shows that 15.4% of 3- to 17-year-olds suffer from overweight, and 5.9% have obesity. There are no differences concerning gender. However, this study shows that children from families with a low socioeconomic status are four times more likely to suffer from severe obesity than their peers who have families with a high socioeconomic status [3,4]. Parents’ health literacy is shown to be closely related to their social status; parents with a higher social status display a higher degree of health literacy than parents with a lower social status. In addition, it is shown that nutrition and the leisure and exercise behaviors of children are decisively influenced by the parents’ health literacy [1,13].

Brandenburg is one of five federal states in Germany in which children suffer the most from overweight and obesity [14]. To assess the situation of students with overweight or obesity in the federal state of Brandenburg, body weight and height measurements were carried out at two different schools (a comprehensive school and a grammar school) in the medium-sized town of Neuruppin, which has a population of about 30,000, in a rural area in the northwest region of Brandenburg. Students of different grade levels were included. Furthermore, a structured questionnaire was distributed.

The aim of the present work was to learn about the local prevalence of obesity and to investigate the interrelation between overweight/obesity and sociopsychological characteristics. In particular, we focused on the following questions: Does the prevalence of overweight and obesity differ from the reported KiGGS results? [3,4]. Are the interrelations between overweight/obesity and socioeconomic status/education level more pronounced in rural areas? What are the psychological effects of overweight and obesity that students experience? 

## 2. Materials and Methods

### 2.1. Study Population

This is a descriptive cross-sectional study. The data were collected at two different schools (a comprehensive school and a grammar school) with assumed educational level differences in different grades (5th, 7th, and 10th grades). The comprehensive school offers general education and concludes with graduation in the 10th grade, whereas the grammar school, referred to as a gymnasium, concludes with graduation in the 12th grade and the receipt of the baccalaureate. This then entitles students to attend a university and pursue a course of study. 

The participating classes were randomly chosen by the respective school administrations. Of the 214 questionnaires, 114 (53%) were answered and returned with the parental consent. The questionnaire response rates and participation in the simultaneous height and weight measurement were 20% in the comprehensive school (25/125) and 100% in the grammar school (89/89). A total of 42 students were attending the 5th grade, 14 students were attending the 7th grade, and 58 students were attending the 10th grade. Of the 114 students, 57 were female and 57 were male. The mean age was 13.2 and ranged from 10 to 18 years. Since most of the children were Caucasian and German, ethnic aspects could be neglected in our evaluation.

This study was approved by the Ethics Committee of the Brandenburg Medical School (MHB, E-01-20181012) and was additionally approved by the State Ministry of Education, Youth and Sport of the Federal State of Brandenburg. Teachers, parents, and students were informed about the study procedures, content, voluntariness, anonymization, and objectives prior to participating. Parental consent was obtained before the data collection process.

### 2.2. Data Collection

In cooperation with the Department of Nutritional Therapy University Hospital Ruppin-Brandenburg, a questionnaire was developed, which is based on the University of Leipzig’s questionnaire titled “The nutritional behavior of adolescents in the context of their lifestyles”. In addition to demographic data, data on nutrition and body weight, on food intake conditions, and on physical and mental well-being were gathered. The students answered the questionnaire independently and anonymously during school lessons.

### 2.3. Determination of BMI

The BMI values of the children and adolescents were measured by 2 people in the afternoon (Ms. Zierenberg and Ms. Schindler). A passenger scale, calibrated with dimensional weights before each measuring unit, was used to determine the weights. For weight measurement, the students took off their jackets and shoes. Height was measured barefoot, using a scale attached to the wall and a right angle to prevent inaccuracies. All measurements were taken in the presence of the teachers. The formula used for calculating the BMI was programmed for adolescents by taking into account age, gender, height, and weight. 

### 2.4. Statistical Analysis

Data from 114 questionnaires were processed and analyzed using SPSS, (IBM SPSS Statistics 20, IBM Corp, Armonk, NY, USA).

Questions that were not answered or answered incorrectly were excluded from the data analysis. These missing values are reported separately in the legends of Figures 1–5. The students’ weights were grouped by applying the percentile curve according to the method used by Kromeyer–Hauschild, which takes into account height, weight, age, and gender [15]. Underweight children or adolescents have a percentile value of <P10, normal-weight children and adolescents have a percentile value of P10–P90, those with pre-obesity have a percentile value of >P90–P97, and those with obesity have a percentile value of *p* > 97. Underweight students were excluded from further analyses due to this study’s design (n = 3).

The parents were assigned to income groups based on their occupations. The average monthly gross salary of both parents in a household was estimated by the Federal Employment Agency and then categorized into three income groups of the same size [16]; these groups were EUR 2215–EUR 5108 for the lower income group, EUR 5134–EUR 6800 for the middle income group, and EUR 6840–EUR 11,596 for the upper income group. The chi-square test was used for categorical variables. Statistical significance was set at *p* < 0.05.

## 3. Results

### 3.1. Demographic Data 

The classification into weight categories using the BMI percentiles according to Kromeyer–Hauschild showed that 76.3% (87/114) of the students had normal weights, 11.4% (13/114) had pre-obesity, and 9.6% (11/114) had obesity. A percentage of 2.6% (3/114) of the students were underweight. The assignment to weight categories did not differ between boys and girls (*p* = 0.242). Next, the students’ body weight statuses were compared according to the type of school. A total of 82% (73/89) of the grammar school students had normal weights, compared to 56% (14/25) of the students attending the comprehensive school. Of the grammar school students, 9% (8/89) had pre-obesity, compared to 20% (5/25) of the comprehensive school students. Children with obesity comprised 6% (5/89) of the students attending grammar school, compared to 24% (6/25) of the comprehensive school students (*p* = 0.009). Overall, 44% (11/25) of the comprehensive school students had pre-obesity or obesity, and for the grammar school students, the percentage was 15% (13/89) (Figure 1). 

There are significant differences in parental income classification concerning the weight categories of the students (*p* = 0.043). For the normal-weight students, 29% (18/62) of the parents fall into the lower income group, 34% (21/62) fall into the middle income group, and 37% (23/62) fall into the upper income group. A total of 67% (6/9) of the parents of students with pre-obesity are in the lower income group, with 22% (2/9) being in the middle income group and 11% (1/9) being in the upper income group. For students with obesity, 38% (3/8) of parents have a lower income, and 63% (5/8) have a middle income. None of the children with obesity in our cohort have parents in the upper income group (Figure 2). 

### 3.2. Nutrition and Weight Perception

More than three-fourths of all students know the food pyramid. The percentage of normal-weight students is 75% (63/84), that of students with pre-obesity is 80% (8/10), and that of students with obesity is 90% (9/10) (*p* = 0.551) [9,15]. An analysis of the answers to eight questions on nutrition and weight perception show significant differences between students of different weight categories for half of the questions. A total of 86% (73/85) of the normal-weight students, 73% (8/11) of the students with pre-obesity, and 9% (1/11) of the students with obesity felt comfortable with their weight (*p* < 0.001). Those who were concerned about their weight included 91% (10/11) of the students with obesity, 36% (4/11) of the students with pre-obesity, and 31% (26/84) of the normal-weight students (*p* < 0.001). At the time of the survey, 27% (26/87) of the normal-weight students, 20% (2/10) of the students with pre-obesity, and 36% (4/11) of the students with obesity (*p* = 0.005) were on a diet. Those who were afraid of gaining weight included 82% (9/11) of the students with obesity, 46% (5/11) of the students with pre-obesity, and 20% (17/86) of the normal-weight children and adolescents (*p* < 0.001) (Figure 3).

### 3.3. Body Perception

A comparison of the students’ body perceptions revealed significant differences. The percentage of students with obesity that felt too overweight is 64% (7/11), while the percentage is 46% (5/11) for the students with pre-obesity, and 8% (7/85) for the normal-weight students (*p* < 0.001). Accordingly, the question of whether one feels exactly right was answered diametrically, with 67% (57/85) for the normal-weight students, 18% (2/11) for the students with pre-obesity, and 9% (1/11) for the students with obesity (*p* < 0.001) (Figure 4).

### 3.4. Importance of Sporting Activities

In principle, 92% (97/105) of all students like to play sports. With regard to school sports, over 80% (83/99) of all students report that they like school sports. However, 14% (11/79) of the normal-weight students, 25% (3/12) of the students with pre-obesity, and 60% (6/10) of the students with obesity reported feeling unsportsmanlike (*p* = 0.002). A total of 6% (5/82) of the normal-weight students, 25% (3/12) of the students with pre-obesity, and 70% (7/10) of the students with obesity feel restricted in performing sporting activities due to their weight (*p* < 0.001). A total of 10% (8/79) of the normal-weight students, 33% (4/12) of the students with pre-obesity, and 70% (7/10) of the students with obesity (*p* < 0.001) reported that they perform sporting activities with the aim of losing weight (Figure 5).

## 4. Discussion 

The aim of our study was to assess the prevalence of overweight and obesity in students in different grades in two schools of a medium-sized Brandenburg city located in a rural area. In addition to assessing the students’ weight categories, parental income was grouped, and the type of school attended (comprehensive or grammar school) was documented. This is particularly interesting, as a strong correlation between parental income and the level of education and type of school attended has previously been reported in Germany [5]. Various components are important for the treatment of obesity in children and adolescents. In addition to diet and exercise, education and behavioral changes are particularly important [17,18,19]. There are numerous studies showing that physical activity has a potential benefit, especially for adolescents with overweight [20,21,22]. Overall, however, the management of obesity, from prevention to treatment, is complex, time-consuming, and costly [23]. Therefore, we were particularly interested in the knowledge of nutrition, the attitude towards eating, body perception, and the importance of sporting activities for students. 

Our study shows that 21% (24/114) of the students had overweight, with 11.4% (13/114) having pre-obesity and 9.6% (11/114) having obesity. The 2018 KiGGS study reported a smaller percentage of 15.4% for children with overweight and 5.9% with obesity between 3 and 17 years of age [3,4]. Therefore, the number of children with overweight was 5.6 percentage points higher in our study population (specifically, 1.9 percentage points for overweight prevalence and 3.7 percentage points higher for obesity prevalence) in comparison with the nationwide KiGSS study (Table 1). The distribution of the weight categories differed significantly with the school type: 44% (11/25) of the students attending the comprehensive school and only 14.6% (13/89) of the students attending the grammar school had pre-obesity or obesity. One reason for this could be the obesogenic impact of school environments in some rural schools, as presented in the Romanian study by Negrea in 2023 [24]. A comparison of our results with those of the RKI study titled “Recognize–Evaluate–Act: On the health of children and adolescents in Germany” from 2008, in which 21.3% of comprehensive school students had overweight (11.3% with pre-obesity and 10% with obesity) and 13.4% of grammar school students had overweight (8.2% with pre-obesity and 5.2% with obesity) (OR: 2.2) shows that the prevalence of overweight in grammar school students has remained almost stable, while the prevalence of overweight for comprehensive school students has doubled [25] (Table 1). 

The fact that this development is even more dramatic in other European countries is shown in a study from Austria involving 8–11-year-old children, with up to 29.2% of them having obesity. The highest increase was recorded during the COVID-19 pandemic in 2020, when children were more inactive [26]. The distribution of students’ weight statuses differed significantly with the parental income grouping. As the parental income increased, the prevalence of obesity among students decreased. In particular, there were no obese children in the upper income group. This finding is in agreement with previously reported data from the KiGGS study group. Children and adolescents of families with a lower socioeconomic status were four times more likely to have obesity than children or adolescents of families with a higher socioeconomic status [3,4]. However, the high number of missing values (approximately 30%), which was similar across all weight categories, has to be taken into account. Social inequality is a sensitive topic, and this might explain the parents’ low willingness to provide information on their occupations to allow for an estimation of their family income. To summarize, our findings strengthen the assumption that socioeconomic disadvantages and fewer educational opportunities are associated with the development of overweight and obesity during childhood and adolescence.

An analysis of the questionnaires enabled us to gain a better understanding of children’s and adolescents’ body perception and attitudes towards nutrition and sports. Interestingly, 75% (63/84) of the normal-weight students, 80% (8/10) of the students with pre-obesity, and 90% (9/10) of the students with obesity reported knowing the food pyramid. Therefore, knowledge of the food pyramid and its use was taught, but the implementation of this knowledge seems to be difficult. More than 90% (97/105) of all students like to play sports in principle [27], with 70% (7/10) of children and adolescents with obesity wanting to play sports with the aim of reducing their body weight [28]. However, 70% (7/10) of the students with obesity feel restricted in performing sporting activities due their body weight. Apparently, the knowledge of healthy nutrition and a positive attitude towards sporting activities are not sufficiently effective for children and adolescents with pre-obesity or obesity to achieve successful body weight regulation. Urbano-Mairena was able to show this phenomenon in Spanish children in 2024, carving out a clear relationship between low physical activity, a high BMI, and lower life satisfaction [29].

The assessment of body perception showed that almost all students with obesity feel uncomfortable with their weight in terms of feeling too corpulent (91%) (10/11), worrying about their weight (91%) (10/11), and being afraid of further weight gain (82%) (9/11). We did not find an underestimation of body circumference, which goes hand in hand with higher satisfaction with body weight, as shown in a Canadian study in 2023 [30]. In contrast, normal-weight students feel largely comfortable with their weight (86%) (73/85), worry less often about their weight (31%) (26/84), and are also less afraid of weight gain (20%) (17/86). These findings are in accordance with the results reported by Jenull et al. (2015) and by Krause et al. (2014) [1,5]. In addition, 20% (2/10) of the students with pre-obesity and 28% (3/11) of the students with obesity reported to be on a diet. These findings highlight the psychological burden of overweight and obesity in children and adolescents. 

A major limitation of our study is the imbalance in participation rates regarding students attending a comprehensive school and a grammar school. The parental consent rate, including the return of the completed questionnaires, was five times lower at the comprehensive school, which was only 20%, compared to that of the grammar school, which was 100%. The significantly different response rates for the questionnaires and the rates of consent could be indications of very different parental attitudes between the two types of schools with regard to education about nutrition and physical activity. 

Therefore, data on the comprehensive school students in particular have to be interpreted with caution and might be biased due to the small sample size. We cannot exclude the fact that there is an over-representation of students with overweight and obesity, nor can we exclude the under-representation. Therefore, further research should focus on assessing overweight and obesity prevalence in comprehensive school students using strategies with the aim of increasing participation. When excluding the question about occupation and parental income, the rate of missing values was 4% in the normal-weight group, 12% in the pre-obese group, and 6% in the obese group, which are within the range of our expectations. 

## 5. Conclusions

With 21% (24/114) of students with pre-obesity and obesity attending two schools in a rural area, there seems to be a higher prevalence of childhood and adolescent overweight and obesity in the federal state of Brandenburg compared to the German average. The attended school forms and parental socioeconomic statuses (derived from the occupational statuses of the parents and estimated parental income) show a significant association with the development of impaired weight management in students. Specifically, attending a grammar school and belonging to an upper parental income group is associated with a decreased prevalence of obesity in children and adolescents. Most of the students know the food pyramid and have positive attitudes towards sporting activities. However, this knowledge and positive attitude towards physical activity do not seem to be sufficiently effective for successful weight regulation. In addition, our study highlights the high burden of overweight and obesity that children and adolescents experience. They feel uncomfortable with their weight, worry about their weight, and fear further weight gain. A better understanding of the psychological stress associated with overweight and obesity and the multifactorial genesis of obesity is necessary to create more targeted interventions to prevent and manage overweight and obesity in childhood and adolescence. To summarize, our work confirms the importance of the parental socioeconomic status and educational opportunities associated with the attended school form in obesity development. Moreover, the considerable burden of overweight and obesity is demonstrated. 

This study was carried out as a part of several scientific internships at the Brandenburg Medical School (MHB) and received no financial support. The authors state that there are no conflicts of interest.

## Figures and Tables

**Figure 1 children-11-00648-f001:**
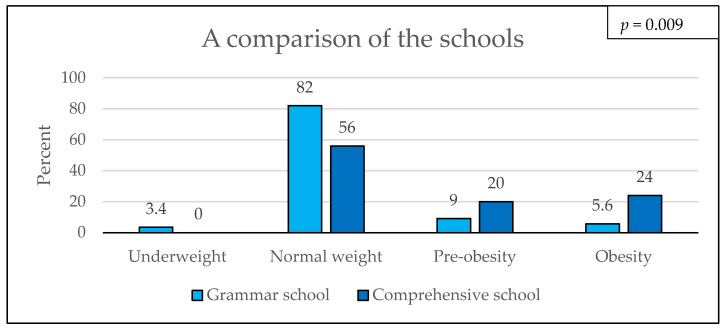
Weight categories according to attendance of grammar school or comprehensive school in percentage; for missing values, percentage: 0%.

**Figure 2 children-11-00648-f002:**
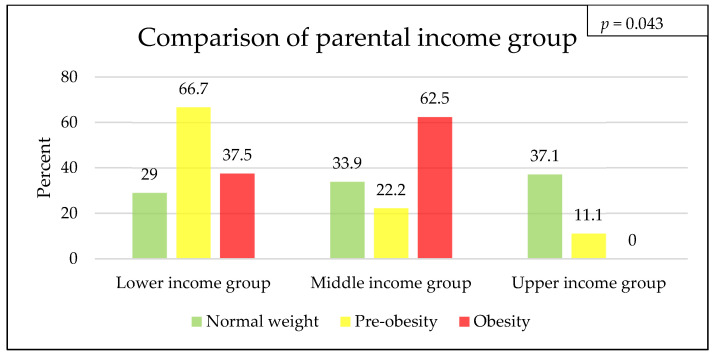
Parental income group according to weight categories, in percentage, and missing values. Normal weight: 25 (28.7%), pre-obese: 4 (30.8%), and obese: 3 (27.8%).

**Figure 3 children-11-00648-f003:**
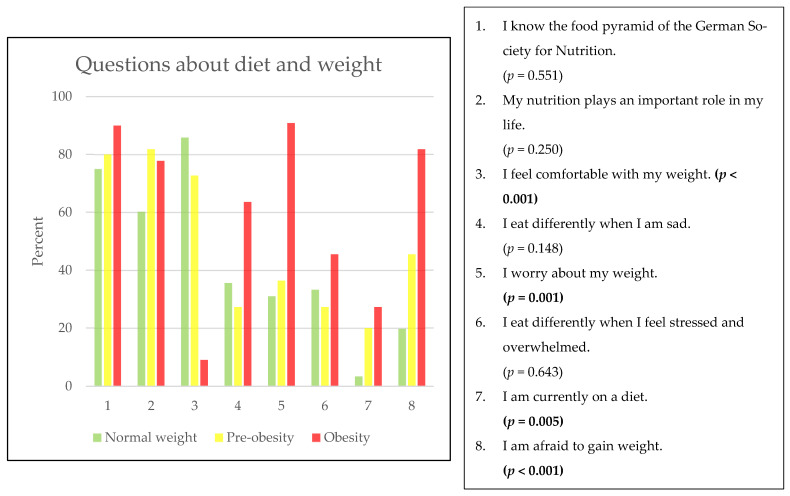
Questions about diet and weight according to weight categories, in %, and missing values. Normal weight: 2 (2.3%), pre-obese: 2 (15.4%), and obese: 1 (9.1%).

**Figure 4 children-11-00648-f004:**
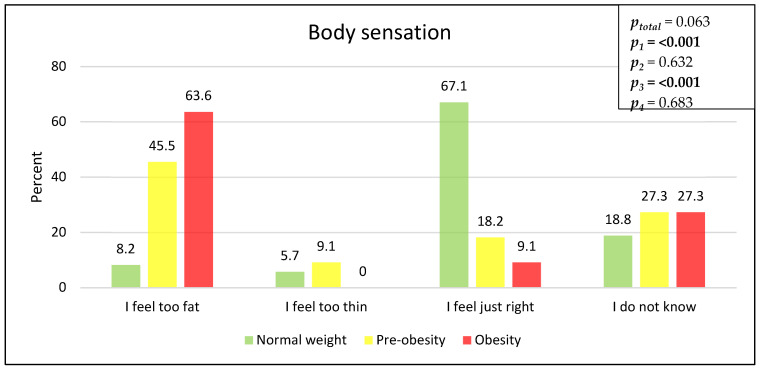
Questions about body perception according to different weight categories, in %, and missing values. Normal weight: 2 (2.3%), pre-obese: 2 (15.4%), and obese: 0.

**Figure 5 children-11-00648-f005:**
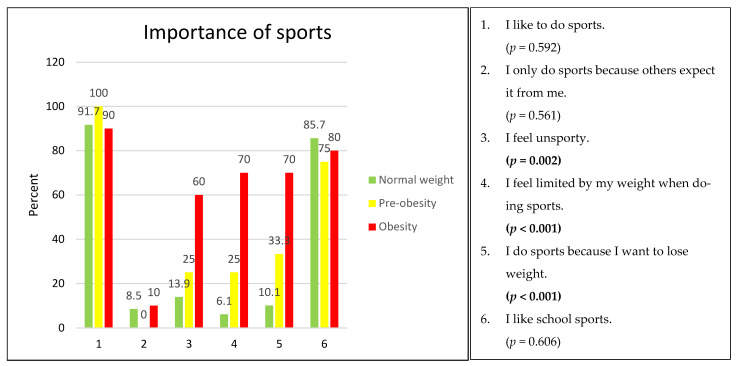
Attitude regarding sporting activities according to weight categories, in %, with missing values. Normal weight: 5 (7.5%), pre-obese: 1 (7.7%), and obese: 1 (9.1%).

**Table 1 children-11-00648-t001:** A comparison of the absolute incidences (%) of overall overweight, pre-obesity, and obesity in schoolchildren in the RKI study and the KIGGS study with the Brandenburg study. The relative percentage values are shown in brackets.

	KIGGS Study [4] (%)	RKI Study [25] (%)	Brandenburg Study (%)
**Pre-obesity**	9.5 (100%)	9.8 (103%)	11.4 (120%)
**Obesity**	5.9 (100%)	7.6 (129%)	9.6 (162%)
**Overall overweight**	15.4 (100%)	17.4 (113%)	21.0 (136%)

## Data Availability

The data are contained within the article.

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
