# Peer review of "Correlation between Obesity and Socioeconomic and Psychological Characteristics of Students Attending Different Rural School Types"

_children, 2024, doi:10.3390/children11060648_

Round 1

Reviewer 1 Report

Comments and Suggestions for Authors

General concept comments:

Children and adolescents with low socioeconomic status (SES) are disproportionally affected by overweight and obesity (Hampl SE, Hassink SG, Skinner AC, et al. Clinical Practice Guideline for the Evaluation and Treatment of Children and Adolescents With Obesity. Pediatrics. 2023;151(2): e2022060640). Understanding familial, cultural and SES is beneficial because of their influence on readiness to change, nutrition factors, access to physical activity and health challenges. Recommendations for pediatric obesity treatment should be integrated within existing community and social systems.

Obesity can also impact children mental health and psychosocial development (American Psychological Association, 2018. Clinical practice guideline for multicomponent behavioral treatment of obesity and overweight in children and adolescents).

The submitted article contributes to understanding the relationship between socioeconomic status (SES) and overweight/obesity among children and adolescents, as well as the investigation of psychological parameters in two different types of schools in a rural region of Germany.

Specific comments:

An article should include the most recent and relevant references in the field. Please update your references.

Abstract: The abstract has 434 words. According to the MPDI instructions for authors the abstract should have 200 words maximum. Adjust your abstract accordingly.

Line 21: Substitute the word “pre-obesity” with overweight. Make this substitution throughout the article (line 129, figure 1, line 157, etc)  or else justify the use of this term.

Lines 21, 22, 97, 160, 161, etc: substitute or justify the use of the words comprehensive and grammar

Line 119: report the types of stadiometer and of scale.

Lines 101-146: Please place the figures in the results section after the paragraph of its first citation.

Line 126: please justify the use of the percentile curve according to Kromeyer- Hauschild. Consider using overweight and obese classification according to the age- and sex-specific definition of the International Obesity Task Force (IOTF) (Cole, T.J.; Lobstein, T. Extended international (IOTF) body mass index cut-offs for thinness, overweight and obesity. Pediatr. Obes. 2012, 7, 284–294.

Lines 210-213. First line treatments of childhood and adolescent obesity are multicomponent interventions including diet, physical activity, educational and behavioral components. Dietary strategies should focus on the promotion of food-based guidelines that target modification of usual eating patterns and behaviors.  Please change your wording accordingly excluding the phrase “hypocaloric diet” and include a reference. Consider the following references 1) EASO and EFAD position statement on Medical Nutrition Therapy for the management of overweight and Obesity in Children and Adolescents, Obesity Facts 2022,  2) Clinical Practice Guideline for the Evaluation and Treatment of Children and Adolescents With Obesity, Pediatrics 2023.

Table 1: replace the phrase “own results” in both table and caption. 

Comments on the Quality of English Language

Please consider editing of English language

Author Response

Comment to the author: An article should include the most recent and relevant references in the field. Please update your references. Response to the reviewer: Dear Reviewer, thank you very much for your valuable work and comments which we appreciated very much and which helped us to improve the manuscript! Thank you very much for hinting to the most recent and relevant references. We included 17 new references in particular, the management of childhood obesity, the guidelines, the role of parents, stigmatization and the psychological problems that characterize children and adolescents who suffer from obesity. The complete reference list now includes 30 references and is presented here: page 10, line 356 – page 12, line 436

  1. Jenull B, Trapp EM. Biopsychosocial approach to childhood and adolescent obesity. Psychotherapist 2015;60(4):337–50.
  2. Demir D, Bektas M. The effect of childrens' eating behaviors and parental feeding style on childhood obesity. Eat Behav. 2017 Aug;26:137-142. doi: 10.1016/j.eatbeh.2017.03.004. Epub 2017 Mar 22. PMID: 28363115.
  3. Hölling H, Kamtsiuris P, Lange M, Thierfelder W, Thamm M, Schlack R. Der Kinder- und Jugendgesundheitssurvey (KiGGS): Studienmanagement und Durchführung der Feldarbeit [The German Health Interview and Examination Survey for Children and Adolescents (KiGGS): study management and conduct of fieldwork]. Bundesgesundheitsblatt Gesundheitsforschung Gesundheitsschutz. 2007 May-Jun;50(5-6):557-66. German. doi: 10.1007/s00103-007-0216-8. PMID: 17514439.
  4. Schienkiewitz A, Brettschneider A-K, Damerow S, Schaffrath Rosario A. Overweight and obesity in childhood and adolescence in Germany - cross-sectional results from KiGGS wave 2 and trends. J Heal Monit. 2018;3(1)
  5. Krause L, Kleiber D, Lampert T. Mental health of overweight and obese adolescents, taking into account social status and school education. Prevention and Health Claim 2014;9(4):264–73.
  6. Wu YK, Berry DC. Impact of weight stigma on physiological and psychological health outcomes for overweight and obese adults: A systematic review. J Adv Nurs. 2018 May;74(5):1030-1042. doi: 10.1111/jan.13511. Epub 2017 Dec 8. PMID: 29171076.
  7. Sutin AR, Stephan Y, Terracciano A. Weight Discrimination and Risk of Mortality. Psychol Sci. 2015 Nov;26(11):1803-11. doi: 10.1177/0956797615601103. Epub 2015 Sep 29. PMID: 26420442; PMCID: PMC4636946.
  8. Jackson SE, Beeken RJ, Wardle J. Perceived weight discrimination and changes in weight, waist circumference, and weight status. Obesity (Silver Spring). 2014 Dec;22(12):2485-8. doi: 10.1002/oby.20891. Epub 2014 Sep 11. PMID: 25212272; PMCID: PMC4236245.
  9. Kirk SF, Price SL, Penney TL, Rehman L, Lyons RF, Piccinini-Vallis H, Vallis TM, Curran J, Aston M. Blame, Shame, and Lack of Support: A Multilevel Study on Obesity Management. Qual Health Res. 2014 Jun;24(6):790-800. doi: 10.1177/1049732314529667. Epub 2014 Apr 11. PMID: 24728109.
  10. Nolan LJ, Eshleman A. Paved with good intentions: Paradoxical eating responses to weight stigma. Appetite. 2016 Jul 1;102:15-24. doi: 10.1016/j.appet.2016.01.027. Epub 2016 Jan 21. PMID: 26802721.
  11. Pearl RL, Puhl RM, Lessard LM, Himmelstein MS, Foster GD. Prevalence and correlates of weight bias internalization in weight management: A multinational study. SSM Popul Health. 2021 Feb 17;13:100755. doi: 10.1016/j.ssmph.2021.100755. PMID: 33718581; PMCID: PMC7920853.
  12. Zorn S. Too fat - without self-esteem - shaped for life: 200,000 adolescents suffer from extreme obesity [Internet]. [cited 2020 Dec 23]. Available from: https://idw-online.de/de/news554787
  13. Shrewsbury VA, Steinbeck KS, Torvaldsen S, Baur LA. The role of parents in pre-adolescent and adolescent overweight and obesity treatment: a systematic review of clinical recommendations. Obes Rev. 2011 Oct;12(10):759-69. doi: 10.1111/j.1467-789X.2011.00882.x. Epub 2011 Apr 27. PMID: 21535361.
  14. Moß A, Wabitsch M, Kromeyer-Hauschild K, Reinehr T, Kurth B-M, Bel Le T. Prevalence of overweight and obesity in German schoolchildren. Bundesgedundheitsblatt 2007;50(11):1424–31.
  15. Kromeyer-Hauschild K, Wabitsch M, Kunze D, et al: Perzentile für den Body-mass-Index für das Kindes- und Jugendalter unter Heranziehung verschiedener deutscher Stichproben. Percentiles of body mass index in children and adolescents evaluated from different regional German cohorts (in German). Monatsschr Kinder- heilkd 2001; 149:807–818.
  16. Remuneration Atlas - Federal Employment Agency [Internet]. [cited 2021 Jan 11]. Available from: https://con.arbeitsagentur.de/prod/entgeltatlas/
  17. Hassapidou M, Duncanson K, Shrewsbury V, Ells L, Mulrooney H, Androutsos O, Vlassopoulos A, Rito A, Farpourt N, Brown T, Douglas P, Ramos Sallas X, Woodward E, Collins C. EASO and EFAD Position Statement on Medical Nutrition Therapy for the Management of Overweight and Obesity in Children and Adolescents. Obes Facts. 2023;16(1):29-52. doi: 10.1159/000527540. Epub 2022 Nov 8. PMID: 36349767; PMCID: PMC9890183.
  18. Hampl SE, Hassink SG, Skinner AC, Armstrong SC, Barlow SE, Bolling CF, Avila Edwards KC, Eneli I, Hamre R, Joseph MM, Lunsford D, Mendonca E, Michalsky MP, Mirza N, Ochoa ER, Sharifi M, Staiano AE, Weedn AE, Flinn SK, Lindros J, Okechukwu K. Clinical Practice Guideline for the Evaluation and Treatment of Children and Adolescents with Obesity. Pediatrics. 2023 Feb 1;151(2):e2022060640. doi: 10.1542/peds.2022-060640. Erratum in: Pediatrics. 2024 Jan 1;153(1): PMID: 36622115.
  19. Vlassopoulos A, Govers E, Mulrooney H, Androutsos O, Hassapidou M. Dietetic management of obesity in Europe: gaps in current practice. Eur J Clin Nutr. 2021 Jul;75(7):1155-1158. doi: 10.1038/s41430-020-00820-2. Epub 2021 Jan 4. PMID: 33398102.
  20. Fanelli E, Abate Daga F, Pappaccogli M, Eula E, Astarita A, Mingrone G, Fasano C, Magnino C, Schiavone D, Rabbone I, Gollin M, Rabbia F, Veglio F. A structured physical activity program in an adolescent population with overweight or obesity: a prospective interventional study. Appl Physiol Nutr Metab. 2022 Mar;47(3):253-260. doi: 10.1139/apnm-2021-0092. Epub 2021 Oct 27. PMID: 34706211.
  21. Schranz N, Tomkinson G, Parletta N, Petkov J, Olds T. Can resistance training change the strength, body composition and self-concept of overweight and obese adolescent males? A randomised controlled trial. Br J Sports Med. 2014 Oct;48(20):1482-8. doi: 10.1136/bjsports-2013-092209. Epub 2013 Aug 14. PMID: 23945035.
  22. Ells LJ, Rees K, Brown T, Mead E, Al-Khudairy L, Azevedo L, McGeechan GJ, Baur L, Loveman E, Clements H, Rayco-Solon P, Farpour-Lambert N, Demaio A. Interventions for treating children and adolescents with overweight and obesity: an overview of Cochrane reviews. Int J Obes (Lond). 2018 Nov;42(11):1823-1833. doi: 10.1038/s41366-018-0230-y. Epub 2018 Oct 9. Erratum in: Int J Obes (Lond). 2019 Apr 2;: PMID: 30301964.
  23. Weihrauch-Blüher S, Kromeyer-Hauschild K, Graf C, Widhalm K, Korsten-Reck U, Jödicke B, Markert J, Müller MJ, Moss A, Wabitsch M, Wiegand S. Current Guidelines for Obesity Prevention in Childhood and Adolescence. Obes Facts. 2018;11(3):263-276. doi: 10.1159/000486512. Epub 2018 Jul 4. PMID: 29969778; PMCID: PMC6103347.
  24. Negrea MO, Negrea GO, Săndulescu G, Neamtu B, Costea RM, Teodoru M, Cipăian CR, Solomon A, Popa ML, Domnariu CD. Assessing Obesogenic School Environments in Sibiu County, Romania: Adapting the ISCOLE School Environment Questionnaire. Children (Basel). 2023 Oct 27;10(11):1746. doi: 10.3390/children10111746. PMID: 38002837; PMCID: PMC10670591.
  25. Recognize – Evaluate – Act: On the health of children and adolescents in Germany. BZgA, Rki. 2008.
  26. Moliterno P, Dornhauser V,Widhalm K Childhood obesity trends among 8-11-year-olds: Insights from a school sample in Vianna, Austria (2017-2023). Children (Basel) 2024 Nov 11, 431. doi 10.3390/children11040431
  27. Mairbäurl H. Body weight and sport. In: Pape H-C, Kurtz A, Silbernagl S, eds. Physiologie: 8., unveränd. Aufl. Stuttgart, Thieme; 2018. pp. 691-692
  28. Count C, Dordel S. Therapy of juvenile obesity from a sports medicine/sports science point of view. Bundesgesundheitsblatt 2011;54(5):541–7.
  29. Urbano-Mairena J, Mendoza-Muñoz M, Carlos-Vivas J, Pastor-Cisneros R, Castillo-Paredes A, Rodal M, Muñoz-Bermejo L. Role of Satisfaction with Life, Sex and Body Mass Index in Physical Literacy of Spanish Children. Children (Basel). 2024 Feb 1;11(2):181. doi: 10.3390/children11020181. PMID: 38397293; PMCID: PMC10886828.
  30. Bordeleau M, Alméras N, Panahi S, Drapeau V. Body Image and Lifestyle Behaviors in High School Adolescents. Children (Basel). 2023 Jul 22;10(7):1263. doi: 10.3390/children10071263. PMID: 37508760; PMCID: PMC10377786.

Comment to the author: Abstract: The abstract has 434 words. According to the MPDI instructions for authors, the abstract should have 200 words maximum. Adjust your abstract accordingly.

Response to the reviewer: Thanks for pointing this out, we have shortened the abstract to 199 words: page 1, line 17-30

Abstract: Weexamined theprevalenceofobesityintwotypesofschools—acomprehensiveschoolandagrammarschool—inaruralGermanregion of Brandenburg.Methods:Inacross-sectionalstudy, BMI values were measured in114studentsingrades5,7,and10.Inadditiontodemographicdata,dataonnutrition,physicalactivity,andmentalwell-being werecollectedusingaquestionnaire.Results:Atotalof44%(11/25)ofcomprehensiveschoolstudentsand15%(13/89)ofhigh school studentsareoverweight, and24%(6/25)ofcomprehensive school  pupilsand6%(5/89)ofgrammar school pupils(p=0.009)areobese.Inaddition,91%(10/11)ofstudents with obesity,36%(4/11)of students with pre-obesity,and31%(26/84)ofnormal-weightstudents(p=0.001)areconcernedabouttheirweight.Amongchildren with obesity,82%(9/11)areafraidofgaining weight. Inaddition,6%(5/82)ofnormal-weight students,25%(3/12)of students withpre-obesity,and70%(7/10)ofstudents with obesityfeelrestrictedbytheirweightwhenexercising.Conclusion:Schoolattendanceandparentalsocioeconomicstatusappear to correlate withstudents'weightstatuses. There is a highlevelofsufferingand theyfeeluncomfortablewiththeirbodiesandworryaboutweightregulation.

Comment to the author: Line 21: Substitute the word “pre-obesity” with overweight. Make this substitution throughout the article (line 129, figure 1, line 157, etc) or else justify the use of this term.

Response to the reviewer: Thank you for this suggestion, we have already dealt with it for a long time when creating the manuscript and would like to keep the classification. Let us explain: The term overweight identifies all persons who have a weight above normal weight and thus represents the overcategory. In this category, however, there are pre-obesitas and obesitas, which in turn differ significantly. If we were to use the term overweight for preobesitas, confusion could occur because obesitas is also one kind of overweight. We have added this exact definition again in the introduction.

Page 1, line 33

  1. Introduction

Prevalence of overweight (pre-obese or obese) is increasing worldwide in childhood and adolescence.

Comment to the author: Lines 21, 22, 97, 160, 161, etc: substitute or justify the use of the words comprehensive and grammar. Response to the reviewer: We have studied 2 types of schools: the general education school with graduation in the 10th grade (comprehensive school) and the gymnasium with graduation in the 12th grade and the receipt of the baccalaureate (grammar school). This then entitles you to attend a university and pursue a course of study. We have included the explanation for all these terms in the methods section.: page 2, line 83-86

The comprehensive school is a general education school with graduation in the 10th grade, and the grammar school is a gymnasium with graduation in the 12th grade and the receipt of the baccalaureate. This then entitles students to attend a university and pursue a course of study.

Comment to the author: Line 119: report the types of stadiometer and of scale. Response to the reviewer: A passenger scale was used. To increase accuracy and comparability, this scale was calibrated with weights prior to measurement. This statement has been added to the methods section: page 3, line 111-118

The BMI values of the children and adolescents were measured by 2 people in the afternoon (Ms. Zierenberg and Ms. Schindler). A passenger scale calibrated with dimensional weights before each measuring unit was used to determine the weights. For weight measurement, the students took off their jackets and shoes. Height was measured barefoot with a scale attached to the wall and a right angle to prevent inaccuracies. All measurements were taken in the presence of the teachers. The formula used for calculating the BMI was programmed for adolescents by taking into account age, gender, height, and weight.

Comment to the author: Lines 101-146: Please place the figures in the results section after the paragraph of its first citation. Response to the reviewer: Thank you, we placed now all figures in the result section after the paragraph of its first citation.

Comment to the author: Line 126: please justify the use of the percentile curve according to Kromeyer- Hauschild. Consider using overweight and obese classification according to the age- and sex-specific definition of the International Obesity Task Force (IOTF) (Cole, T.J.; Lobstein, T. Extended international (IOTF) body mass index cut-offs for thinness, overweight and obesity. Pediatr. Obes. 2012, 7, 284–294. Response to the reviewer: Thanks very much for pointing this out. We would like to explain the reason for the chosen method. The pupils were grouped into the different weight classes according to the percentile curves to Kromeyer and Hausschild, which are common in Germany. In addition to height and weight, these also take into account age and gender in children, as this has a significant influence on children. We believe that this is a very precise assignment to the individual weight classes. Furthermore, we have now included the exact literature on this. Page 11, line 391-393

Kromeyer-Hauschild K, Wabitsch M, Kunze D, et al: Perzentile für den Body-mass-Index für das Kindes- und Jugendalter unter Heranziehung verschiedener deutscher Stichproben. Percentiles of body mass index in children and adolescents evaluated from different regional German cohorts (in German). Monatsschr Kinder- heilkd 2001; 149:807–818.

Comment to the author: Lines 210-213. First line treatments of childhood and adolescent obesity are multicomponent interventions including diet, physical activity, educational and behavioral components. Dietary strategies should focus on the promotion of food-based guidelines that target modification of usual eating patterns and behaviors.  Please change your wording accordingly excluding the phrase “hypocaloric diet” and include a reference. Consider the following references 1) EASO and EFAD position statement on Medical Nutrition Therapy for the management of overweight and Obesity in Children and Adolescents, Obesity Facts 2022,  2) Clinical Practice Guideline for the Evaluation and Treatment of Children and Adolescents With Obesity, Pediatrics 2023. Response to the reviewer: Thank you very much for the valuable hint. We are aware that, in addition to nutrition and exercise components, educational and behavioral components are also needed for the obesity treatment of children. That's why we've changed the rate from in the section discussion as follows. Page 7, line 228-232

Various components are important for the treatment of obesity in children and adolescents. In addition to diet and exercise, education and behavioral changes are particularly important (17, 18, 19). There are numerous studies showing that physical activity has a potential benefit, especially for adolescents with overweight (20, 21, 22). Overall, however, the management of obesity from prevention to treatment is complex, time-consuming, and costly (23).

In this regard, we have included the following literature sources. Page 11, line 396-420

  1. Hassapidou M, Duncanson K, Shrewsbury V, Ells L, Mulrooney H, Androutsos O, Vlassopoulos A, Rito A, Farpourt N, Brown T, Douglas P, Ramos Sallas X, Woodward E, Collins C. EASO and EFAD Position Statement on Medical Nutrition Therapy for the Management of Overweight and Obesity in Children and Adolescents. Obes Facts. 2023;16(1):29-52. doi: 10.1159/000527540. Epub 2022 Nov 8. PMID: 36349767; PMCID: PMC9890183.
  2. Hampl SE, Hassink SG, Skinner AC, Armstrong SC, Barlow SE, Bolling CF, Avila Edwards KC, Eneli I, Hamre R, Joseph MM, Lunsford D, Mendonca E, Michalsky MP, Mirza N, Ochoa ER, Sharifi M, Staiano AE, Weedn AE, Flinn SK, Lindros J, Okechukwu K. Clinical Practice Guideline for the Evaluation and Treatment of Children and Adolescents with Obesity. Pediatrics. 2023 Feb 1;151(2):e2022060640. doi: 10.1542/peds.2022-060640. Erratum in: Pediatrics. 2024 Jan 1;153(1): PMID: 36622115.
  3. Vlassopoulos A, Govers E, Mulrooney H, Androutsos O, Hassapidou M. Dietetic management of obesity in Europe: gaps in current practice. Eur J Clin Nutr. 2021 Jul;75(7):1155-1158. doi: 10.1038/s41430-020-00820-2. Epub 2021 Jan 4. PMID: 33398102.
  4. Fanelli E, Abate Daga F, Pappaccogli M, Eula E, Astarita A, Mingrone G, Fasano C, Magnino C, Schiavone D, Rabbone I, Gollin M, Rabbia F, Veglio F. A structured physical activity program in an adolescent population with overweight or obesity: a prospective interventional study. Appl Physiol Nutr Metab. 2022 Mar;47(3):253-260. doi: 10.1139/apnm-2021-0092. Epub 2021 Oct 27. PMID: 34706211.
  5. Schranz N, Tomkinson G, Parletta N, Petkov J, Olds T. Can resistance training change the strength, body composition and self-concept of overweight and obese adolescent males? A randomised controlled trial. Br J Sports Med. 2014 Oct;48(20):1482-8. doi: 10.1136/bjsports-2013-092209. Epub 2013 Aug 14. PMID: 23945035.
  6. Ells LJ, Rees K, Brown T, Mead E, Al-Khudairy L, Azevedo L, McGeechan GJ, Baur L, Loveman E, Clements H, Rayco-Solon P, Farpour-Lambert N, Demaio A. Interventions for treating children and adolescents with overweight and obesity: an overview of Cochrane reviews. Int J Obes (Lond). 2018 Nov;42(11):1823-1833. doi: 10.1038/s41366-018-0230-y. Epub 2018 Oct 9. Erratum in: Int J Obes (Lond). 2019 Apr 2;: PMID: 30301964.
  7. Weihrauch-Blüher S, Kromeyer-Hauschild K, Graf C, Widhalm K, Korsten-Reck U, Jödicke B, Markert J, Müller MJ, Moss A, Wabitsch M, Wiegand S. Current Guidelines for Obesity Prevention in Childhood and Adolescence. Obes Facts. 2018;11(3):263-276. doi: 10.1159/000486512. Epub 2018 Jul 4. PMID: 29969778; PMCID: PMC6103347.

Comment to the author: Table 1: replace the phrase “own results” in both table and caption. 
Response to the reviewer: We're not sure if it was just the term "own results" that was unfortunate for us, or if we should delete the entire column. Since the table shows well in comparison with the KIGGS and the RKI-study that our numbers continue to rise in all 3 categories, we would leave the table as it is if possible and change the heading to Brandenburg-study. Page 8, line 254

Table 1. Comparison of the absolute incidences (%) of overall overweight, pre-obesity and obesity of school children of the RKI-study and the KIGGS-study with Brandenburg-study. Relative Percentage values in brackets.

KIGGS -study (2)

(%)

RKI-study (9)

(%)

Brandenburg-study (%)

Pre-obese

9,5 (100%)

9,8 (103%)

11,4 (120%)

Obese

5,9 (100%)

7,6 (129%)

 9,6 (162%)

Overall overweight

15,4 (100%)

17,4 (113%)

21,0 (136%)

Comment to the author: Please consider editing of English language Response to the reviewer: We used the author service from MDPI for English editing.

Comment to the author: (Introduction) In line 70, you mentioned “The nationwide KiGGS (study on the health of children, adolescents and young 70 adults in Germany).” I suggest you insert the German acronym before its English translation. This may help the reader match the capital letters with the proper German word (even if they do not know German, but it maintains coherence while reading). Response to the reviewer: As suggested, we have added the German meaning for KiGGS: "German Child and Youth Health Survey" and at the same time indicated the corresponding original literature. Page 2, line 54-57

The nationwide KiGGS (German Child and Youth Health Survey) is a study that focuses on the health of children, adolescents, and young adults at the Robert Koch Institute in Germany, covering the years of 2014 to 2017. It shows that 15.4% of 3- to 17-year-olds suffer from overweight, and 5.9% have obesity.

Page 10, line 360-363

Hölling H, Kamtsiuris P, Lange M, Thierfelder W, Thamm M, Schlack R. Der Kinder- und Jugendgesundheitssurvey (KiGGS): Studienmanagement und Durchführung der Feldarbeit [The German Health Interview and Examination Survey for Children and Adolescents (KiGGS): study management and conduct of fieldwork]. Bundesgesundheitsblatt Gesundheitsforschung Gesundheitsschutz. 2007 May-Jun;50(5-6):557-66. German. doi: 10.1007/s00103-007-0216-8. PMID: 17514439.

Reviewer 2 Report

Comments and Suggestions for Authors

Dear authors,

Thank you for your manuscript. This article has good potential and may have an excellent scientific

impact. In this direction, I have some advice for you:

ABSTRACT:

Abstract is okay

INTRODUCTION:

In line 70, you mentioned “The nationwide KiGGS (study on the health of children, adolescents and

young 70 adults in Germany).” I suggest you insert the German acronym before its English

translation. This may help the reader match the capital letters with the proper German word (even if

they do not know German, but it maintains coherence while reading).

I suggest you write a very short paragraph that briefly explains how the German school system works

during childhood. For example, I cannot focus on the difference between grammar school and

comprehensive school in the method section. If this will take too long, please explain just the

difference in the method section.

METHODS

BMI with which formula?

Please describe the measurement procedure. Was it taken in the morning or in the afternoon? In the

school or in a medical centre? How was it registered? With a stadiometer? With a meter fixed on the

wall? With which model of scale? Please provide full information about this procedure to help the

reader understand your study better.

RESULTS

The result section is generally okay, but have you considered including the children’s ethnic

background in the analysis? If yes, please provide data. If not, please explain why (maybe simply

affirming that most children were Caucasian and German). Ethnic background can be a bias for

obesity monitoring because of cultural heritage.

DISCUSSION

In line 100 of the method section, you declared, “The questionnaire response 100 rate and

simultaneous measurement of weight and height was 20% in the comprehensive school (25/125)

and 100% in the grammar school (89/89)”. Please discuss this outcome in the discussion section.

CONCLUSION

It is okay

REFERENCES

Must be improved. At least 22-23 references.

Add a little paragraph about the potential benefits of physical activity and sport participation in

reducing obesity in you introduction section to improve your references. Here there are two potential

studies that can help you improve your bibliografy or find other studies directly focused on your topic.

Fanelli E, Abate Daga F, Pappaccogli M, Eula E, Astarita A, Mingrone G, Fasano C, Magnino C,

Schiavone D, Rabbone I, Gollin M, Rabbia F, Veglio F. A structured physical activity program in an

adolescent population with overweight or obesity: a prospective interventional study. Appl Physiol

Nutr Metab. 2022 Mar;47(3):253-260. doi: 10.1139/apnm-2021-0092. Epub 2021 Oct 27. PMID:

34706211.

Schranz, G. Tomkinson, N. Parletta, J. Petkov, and T. Olds, “Can resistance training change the

strength, body composition and self-concept of overweight and obese adolescent males? A

randomised controlled trial,” Br. J. Sports Med., vol. 48, no. 20, pp. 1482–1488, 201

Thank you for the opportunity to revise this article. I hope that my suggestions will be helpful for your

manuscript.

Author Response

Comment to the author (Introduction) I suggest you write a very short paragraph that briefly explains how the German school system works during childhood. For example, I cannot focus on the difference between grammar school and comprehensive school in the method section. If this will take too long, please explain just the difference in the method section. Response to the reviewer: We have studied 2 types of schools: the general education school with graduation in the 10th grade (comprehensive school) and the gymnasium with graduation in the 12th grade and the receipt of the baccalaureate (grammar school). This then entitles you to attend a university and pursue a course of study. We have included the explanation for all these terms in the methods section.:page 2, line 83-86

The comprehensive school is a general education school with graduation in the 10th grade, and the grammar school is a gymnasium with graduation in the 12th grade and the receipt of the baccalaureate. This then entitles students to attend a university and pursue a course of study.

Comment to the author (Methods) BMI with which formula? Please describe the measurement procedure. Was it taken in the morning or in the afternoon? In the school or in a medical center? How was it registered? With a stadiometer? With a meter fixed on the wall? With which model of scale? Please provide full information about this procedure to help the reader understand your study better. Response to the reviewer: Thank you very much for the valuable hint. Accordingly, we rewrite the section method: page 3, line 111-118

2.3. Determination BMI

The BMI values of the children and adolescents were measured by 2 people in the afternoon (Ms. Zierenberg and Ms. Schindler). A passenger scale calibrated with dimensional weights before each measuring unit was used to determine the weights. For weight measurement, the students took off their jackets and shoes. Height was measured barefoot with a scale attached to the wall and a right angle to prevent inaccuracies. All measurements were taken in the presence of the teachers. The formula used for calculating the BMI was programmed for adolescents by taking into account age, gender, height, and weight.

Comment to the author (Results) The result section is generally okay, but have you considered including the children’s ethnic background in the analysis? If yes, please provide data. If not, please explain why (maybe simply affirming that most children were Caucasian and German). Ethnic background can be a bias for obesity monitoring because of cultural heritage. Response to the reviewer: Thank you very much for the inquiry. We did not have to deal with ethnic aspects, as the vast majority of children were Caucasian and German. We have included this in the study population and methods section as follows: page 2, line 94-page 2, line 95

Since most of the children were Caucasian and German, ethnic aspects could be neglected in our evaluation.

Comment to the author:(Discussion) In line 100 of the method section, you declared, “The questionnaire response rate and simultaneous measurement of weight and height was 20% in the. comprehensive school (25/125) and 100% in the grammar school (89/89)”. Please discuss this outcome in the discussion section. Response to the reviewer: You are absolutely right that this fact needs to be discussed. Of the questionnaires and consent forms originally issued, we received a total of 53% (114) back. Of course, we were only able to examine these. In general, such a response rate is not unusual, but if we look at the two types of school individually, the interest in participating in such a study is significantly lower for the comprehensive school with a 20% response rate than for the grammar school with 100%. Accordingly, we added the following in the discussion: page 9, line 301-303

The significantly different response rates for the questionnaires and the rates of consent could be indications of very different parental attitudes between the two types of schools with regard to education about nutrition and physical activity.

Comment to the author:(References) Must be improved. At least 22-23 references. Add a little paragraph about the potential benefits of physical activity and sport participation in reducing obesity in you introduction section to improve your references. Here there are two potential studies that can help you improve your bibliografy or find other studies directly focused on your topic. Fanelli E, Abate Daga F, Pappaccogli M, Eula E, Astarita A, Mingrone G, Fasano C, Magnino C, Schiavone D, Rabbone I, Gollin M, Rabbia F, Veglio F. A structured physical activity program in an adolescent population with overweight or obesity: a prospective interventional study. Appl Physiol Nutr Metab. 2022 Mar;47(3):253-260. doi: 10.1139/apnm-2021-0092. Epub 2021 Oct 27. PMID:34706211. Schranz, G. Tomkinson, N. Parletta, J. Petkov, and T. Olds, “Can resistance training change the strength, body composition and self-concept of overweight and obese adolescent males? A randomised controlled trial,” Br. J. Sports Med., vol. 48, no. 20, pp. 1482–1488, 201 Response to the reviewer: Thank you very much for the kind and valuable hint. We have included in the discussion (it is better to fit in) a short section on the importance of physical activity. At the same time, we added 17 new references and thus in the end 30 references. Page 7, line 228-232

Various components are important for the treatment of obesity in children and adolescents. In addition to diet and exercise, education and behavioral changes are particularly important (17, 18, 19). There are numerous studies showing that physical activity has a potential benefit, especially for adolescents with overweight (20, 21, 22). Overall, however, the management of obesity from prevention to treatment is complex, time-consuming, and costly (23).

page 10, line 356-436

  1. Jenull B, Trapp EM. Biopsychosocial approach to childhood and adolescent obesity. Psychotherapist 2015;60(4):337–50.
  2. Demir D, Bektas M. The effect of childrens' eating behaviors and parental feeding style on childhood obesity. Eat Behav. 2017 Aug;26:137-142. doi: 10.1016/j.eatbeh.2017.03.004. Epub 2017 Mar 22. PMID: 28363115.
  3. Hölling H, Kamtsiuris P, Lange M, Thierfelder W, Thamm M, Schlack R. Der Kinder- und Jugendgesundheitssurvey (KiGGS): Studienmanagement und Durchführung der Feldarbeit [The German Health Interview and Examination Survey for Children and Adolescents (KiGGS): study management and conduct of fieldwork]. Bundesgesundheitsblatt Gesundheitsforschung Gesundheitsschutz. 2007 May-Jun;50(5-6):557-66. German. doi: 10.1007/s00103-007-0216-8. PMID: 17514439.
  4. Schienkiewitz A, Brettschneider A-K, Damerow S, Schaffrath Rosario A. Overweight and obesity in childhood and adolescence in Germany - cross-sectional results from KiGGS wave 2 and trends. J Heal Monit. 2018;3(1)
  5. Krause L, Kleiber D, Lampert T. Mental health of overweight and obese adolescents, taking into account social status and school education. Prevention and Health Claim 2014;9(4):264–73.
  6. Wu YK, Berry DC. Impact of weight stigma on physiological and psychological health outcomes for overweight and obese adults: A systematic review. J Adv Nurs. 2018 May;74(5):1030-1042. doi: 10.1111/jan.13511. Epub 2017 Dec 8. PMID: 29171076.
  7. Sutin AR, Stephan Y, Terracciano A. Weight Discrimination and Risk of Mortality. Psychol Sci. 2015 Nov;26(11):1803-11. doi: 10.1177/0956797615601103. Epub 2015 Sep 29. PMID: 26420442; PMCID: PMC4636946.
  8. Jackson SE, Beeken RJ, Wardle J. Perceived weight discrimination and changes in weight, waist circumference, and weight status. Obesity (Silver Spring). 2014 Dec;22(12):2485-8. doi: 10.1002/oby.20891. Epub 2014 Sep 11. PMID: 25212272; PMCID: PMC4236245.
  9. Kirk SF, Price SL, Penney TL, Rehman L, Lyons RF, Piccinini-Vallis H, Vallis TM, Curran J, Aston M. Blame, Shame, and Lack of Support: A Multilevel Study on Obesity Management. Qual Health Res. 2014 Jun;24(6):790-800. doi: 10.1177/1049732314529667. Epub 2014 Apr 11. PMID: 24728109.
  10. Nolan LJ, Eshleman A. Paved with good intentions: Paradoxical eating responses to weight stigma. Appetite. 2016 Jul 1;102:15-24. doi: 10.1016/j.appet.2016.01.027. Epub 2016 Jan 21. PMID: 26802721.
  11. Pearl RL, Puhl RM, Lessard LM, Himmelstein MS, Foster GD. Prevalence and correlates of weight bias internalization in weight management: A multinational study. SSM Popul Health. 2021 Feb 17;13:100755. doi: 10.1016/j.ssmph.2021.100755. PMID: 33718581; PMCID: PMC7920853.
  12. Zorn S. Too fat - without self-esteem - shaped for life: 200,000 adolescents suffer from extreme obesity [Internet]. [cited 2020 Dec 23]. Available from: https://idw-online.de/de/news554787
  13. Shrewsbury VA, Steinbeck KS, Torvaldsen S, Baur LA. The role of parents in pre-adolescent and adolescent overweight and obesity treatment: a systematic review of clinical recommendations. Obes Rev. 2011 Oct;12(10):759-69. doi: 10.1111/j.1467-789X.2011.00882.x. Epub 2011 Apr 27. PMID: 21535361.
  14. Moß A, Wabitsch M, Kromeyer-Hauschild K, Reinehr T, Kurth B-M, Bel Le T. Prevalence of overweight and obesity in German schoolchildren. Bundesgedundheitsblatt 2007;50(11):1424–31.
  15. Kromeyer-Hauschild K, Wabitsch M, Kunze D, et al: Perzentile für den Body-mass-Index für das Kindes- und Jugendalter unter Heranziehung verschiedener deutscher Stichproben. Percentiles of body mass index in children and adolescents evaluated from different regional German cohorts (in German). Monatsschr Kinder- heilkd 2001; 149:807–818.
  16. Remuneration Atlas - Federal Employment Agency [Internet]. [cited 2021 Jan 11]. Available from: https://con.arbeitsagentur.de/prod/entgeltatlas/
  17. Hassapidou M, Duncanson K, Shrewsbury V, Ells L, Mulrooney H, Androutsos O, Vlassopoulos A, Rito A, Farpourt N, Brown T, Douglas P, Ramos Sallas X, Woodward E, Collins C. EASO and EFAD Position Statement on Medical Nutrition Therapy for the Management of Overweight and Obesity in Children and Adolescents. Obes Facts. 2023;16(1):29-52. doi: 10.1159/000527540. Epub 2022 Nov 8. PMID: 36349767; PMCID: PMC9890183.
  18. Hampl SE, Hassink SG, Skinner AC, Armstrong SC, Barlow SE, Bolling CF, Avila Edwards KC, Eneli I, Hamre R, Joseph MM, Lunsford D, Mendonca E, Michalsky MP, Mirza N, Ochoa ER, Sharifi M, Staiano AE, Weedn AE, Flinn SK, Lindros J, Okechukwu K. Clinical Practice Guideline for the Evaluation and Treatment of Children and Adolescents with Obesity. Pediatrics. 2023 Feb 1;151(2):e2022060640. doi: 10.1542/peds.2022-060640. Erratum in: Pediatrics. 2024 Jan 1;153(1): PMID: 36622115.
  19. Vlassopoulos A, Govers E, Mulrooney H, Androutsos O, Hassapidou M. Dietetic management of obesity in Europe: gaps in current practice. Eur J Clin Nutr. 2021 Jul;75(7):1155-1158. doi: 10.1038/s41430-020-00820-2. Epub 2021 Jan 4. PMID: 33398102.
  20. Fanelli E, Abate Daga F, Pappaccogli M, Eula E, Astarita A, Mingrone G, Fasano C, Magnino C, Schiavone D, Rabbone I, Gollin M, Rabbia F, Veglio F. A structured physical activity program in an adolescent population with overweight or obesity: a prospective interventional study. Appl Physiol Nutr Metab. 2022 Mar;47(3):253-260. doi: 10.1139/apnm-2021-0092. Epub 2021 Oct 27. PMID: 34706211.
  21. Schranz N, Tomkinson G, Parletta N, Petkov J, Olds T. Can resistance training change the strength, body composition and self-concept of overweight and obese adolescent males? A randomised controlled trial. Br J Sports Med. 2014 Oct;48(20):1482-8. doi: 10.1136/bjsports-2013-092209. Epub 2013 Aug 14. PMID: 23945035.
  22. Ells LJ, Rees K, Brown T, Mead E, Al-Khudairy L, Azevedo L, McGeechan GJ, Baur L, Loveman E, Clements H, Rayco-Solon P, Farpour-Lambert N, Demaio A. Interventions for treating children and adolescents with overweight and obesity: an overview of Cochrane reviews. Int J Obes (Lond). 2018 Nov;42(11):1823-1833. doi: 10.1038/s41366-018-0230-y. Epub 2018 Oct 9. Erratum in: Int J Obes (Lond). 2019 Apr 2;: PMID: 30301964.
  23. Weihrauch-Blüher S, Kromeyer-Hauschild K, Graf C, Widhalm K, Korsten-Reck U, Jödicke B, Markert J, Müller MJ, Moss A, Wabitsch M, Wiegand S. Current Guidelines for Obesity Prevention in Childhood and Adolescence. Obes Facts. 2018;11(3):263-276. doi: 10.1159/000486512. Epub 2018 Jul 4. PMID: 29969778; PMCID: PMC6103347.
  24. Negrea MO, Negrea GO, Săndulescu G, Neamtu B, Costea RM, Teodoru M, Cipăian CR, Solomon A, Popa ML, Domnariu CD. Assessing Obesogenic School Environments in Sibiu County, Romania: Adapting the ISCOLE School Environment Questionnaire. Children (Basel). 2023 Oct 27;10(11):1746. doi: 10.3390/children10111746. PMID: 38002837; PMCID: PMC10670591.
  25. Recognize – Evaluate – Act: On the health of children and adolescents in Germany. BZgA, Rki. 2008.
  26. Moliterno P, Dornhauser V,Widhalm K Childhood obesity trends among 8-11-year-olds: Insights from a school sample in Vianna, Austria (2017-2023). Children (Basel) 2024 Nov 11, 431. doi 10.3390/children11040431
  27. Mairbäurl H. Body weight and sport. In: Pape H-C, Kurtz A, Silbernagl S, eds. Physiologie: 8., unveränd. Aufl. Stuttgart, Thieme; 2018. pp. 691-692
  28. Count C, Dordel S. Therapy of juvenile obesity from a sports medicine/sports science point of view. Bundesgesundheitsblatt 2011;54(5):541–7.
  29. Urbano-Mairena J, Mendoza-Muñoz M, Carlos-Vivas J, Pastor-Cisneros R, Castillo-Paredes A, Rodal M, Muñoz-Bermejo L. Role of Satisfaction with Life, Sex and Body Mass Index in Physical Literacy of Spanish Children. Children (Basel). 2024 Feb 1;11(2):181. doi: 10.3390/children11020181. PMID: 38397293; PMCID: PMC10886828.
  30. Bordeleau M, Alméras N, Panahi S, Drapeau V. Body Image and Lifestyle Behaviors in High School Adolescents. Children (Basel). 2023 Jul 22;10(7):1263. doi: 10.3390/children10071263. PMID: 37508760; PMCID: PMC10377786.

Round 2

Reviewer 2 Report

Comments and Suggestions for Authors

Dear authors,

Thank you for the corrections and improvements to your manuscript. It is much more powerful now than before.  Congratulations on your job!